# ADSS: Boosting Text-to-Image Diffusion Models via Attention-Driven Seed Selection

## Abstract

Text-to-image diffusion models can synthesize high-quality images, yet the outcome is notoriously sensitive to the random seed: different initial seeds often yield large variations in image quality and prompt–image alignment. We revisit this "seed effect" and show that early-stage attention dynamics over prompt core tokens—the content-bearing words—strongly predict final generation quality. Building on this observation, we introduce ADSS—Attention-Driven Seed Selection—a training-free, plug-and-play method that tracks cross-attention to core tokens during sampling to rank and select seeds for a fixed prompt, requiring no finetuning or latent changes and globally ranking the entire seed pool rather than using a fixed threshold. Since it operates purely at inference time, ADSS can also serve as a lightweight add-on preselection step before existing seed-optimization pipelines, enabling additional gains without extra training or code changes. Extensive experiments on three benchmarks show consistent improvements in prompt faithfulness and visual quality across Stable Diffusion variants, as reflected by human preference and alignment metrics. Our results highlight ADSS as a simple and effective route to more controllable generation by leveraging prompt core token attention for robust seed preselection.

## 1 Introduction

Text-to-Image Synthesis (T2I) has rapidly evolved into a central topic in generative modeling, aiming to produce realistic and semantically consistent images directly from natural language prompts. Early approaches widely leveraged text-conditioned generative adversarial networks (Reed et al., 2016; Zhang et al., 2017; Xu et al., 2018), conditional variational autoencoders (Sohn et al., 2015), and autoregressive models (Ramesh et al., 2021) to improve semantic alignment and diversity of generated images. Diffusion models (Ho et al., 2020; Song et al., 2021b; Liu et al., 2024a) have driven a paradigm shift in generative modeling, establishing themselves as the leading approach through their stable training process and exceptional output quality. In particular, combining diffusion models with large-scale language/vision–language representations has driven major breakthroughs in T2I synthesis (Nichol et al., 2022; Ramesh et al., 2022; Saharia et al., 2022; Rombach et al., 2022; Podell et al., 2023; Balaji et al., 2022).

Despite recent advances, diffusion-based T2I systems remain highly sensitive to the seed-defined initial latent, often yielding noticeably variable results. Even small perturbations to this initialization can significantly affect output fidelity, aesthetics, and semantic alignment (Mao et al., 2023; Li et al., 2025). To mitigate this issue, prior work modifies the initial random latent determined by seed to improve generation quality. We refer to studies in this line of research as seed optimization. Approaches include reward-based optimization with preference-guided objectives (Eyring et al., 2024; Miao et al., 2024), attention-guided refinement via cross-/self-attention control (Hong et al., 2023; Chefer et al., 2023; Guo et al., 2024), and controllable rollback techniques using inversion-based backtracking (Bai et al., 2024; Mao et al., 2025; Qi et al., 2024). In contrast, seed selection operates over a large pool of random seeds and focuses on identifying a subset that produces consistently high-quality outputs. For example, Xu et al. (2024b) introduced the concept of Golden Seeds by running extensive generation trials on validation set and selecting seeds that consistently yield superior results, which are then applied to the target set. Similarly, Li et al. (2025) identified high-performing seeds from generated datasets and reused them for fine-tuning a frozen diffusion model.

**Contribution**. In this paper, we focus on the seed selection problem. Unlike existing seed selection methods that rely on auxiliary datasets, which incurs additional seed curation/fine-tuning costs , our approach distinguishes between good and bad seeds[1] during the denoising process itself, particularly in the early stages. Our key contributions can be summarized as follows:

- We propose ADSS (Attention-Driven Seed Selection) — a training-free, plug-and-play procedure for selecting seeds at inference time, leveraging the denoising process itself without requiring any external supervision.
- We reveal a key insight: early-stage cross-attention on core tokens is a strong predictor of final prompt alignment and image quality, providing the base for deriving a simple yet effective criterion for screening and early stopping of low-quality seeds.
- We validate ADSS through extensive experiments on three benchmarks and multiple Stable Diffusion variants, demonstrating consistent gains, by widely used metrics in both semantic alignment and correlation with human judgments..

## 2    RELATED WORK

**Text-to-Image Diffusion Models**. T2I diffusion models have rapidly become one of the most powerful generative model families, capable of synthesizing highly diverse and photorealistic images conditioned on natural language descriptions (Saharia et al., 2022; Rombach et al., 2022). These models typically incorporate pretrained language encoders—such as CLIP (Radford et al., 2021), T5 (Raffel et al., 2020), or more recently large language models (Balaji et al., 2022)—to transform textual inputs into dense representations. The encoded information is then injected into the generative process via cross-attention layers, which serve as the primary mechanism for aligning semantics between text and image (Vaswani et al., 2017; Rombach et al., 2022). While this architecture has achieved remarkable success in controllable generation, the outputs can be highly sensitive to the initial random seeds, leading to significant quality disparity between good and bad seeds; typical failures include missing objects/parts, inappropriate overlaps or misplacements, and even spurious text artifacts in inpainting (Xu et al., 2024b; Shen et al., 2025). Moreover, T2I models are known to suffer compositional errors—e.g., incorrect counts, positions, and attribute binding—yet recent evidence shows that a substantial portion of these failures is in fact seed-dependent (Li et al., 2025; Gokhale et al., 2023; Liu et al., 2024b; Ban et al., 2025).

**Seed Optimization in Text-to-Image Diffusion Models**. Recent work aims to remedy inferior generations by optimizing the seed-defined initial latent. A prominent line of research (Hertz et al., 2023) pursues attention-guided refinement, leveraging cross-attention maps to better align images with prompts—tokens receiving higher attention are encouraged to be more strongly expressed. Attend-and-Excite introduces a normalized attention loss that iteratively updates the latent to mitigate subject neglect (Chefer et al., 2023). Follow-ups further reweight token attention or enforce coverage/consistency—without retraining—to reduce seed-specific failure modes (Agarwal et al., 2023; Rassin et al., 2023; Meral et al., 2024; Guo et al., 2024; Qiu et al., 2025).

Complementary to attention guidance, Z-sampling (Bai et al., 2024; Mao et al., 2025) mitigates inference-time suboptimality by inserting controlled "back" moves—partial inversion or noise re-injection—into the denoising trajectory to escape poor basins; Ctrl-Z Sampling invokes these rollbacks under a reward signal before continuing refinement. Crucially, each rollback-retrack cycle reconditions the trajectory with additional signals (e.g., refreshed attention/saliency or reward feedback), effectively increasing the informational conditioning available to the sampler and stabilizing quality across seeds. Relatedly, Golden Noise (Zhou et al., 2024) learns a prompt-conditioned "noise prompt" that transforms a random seed into a golden seed via a lightweight NPNet, improving alignment and aesthetics across various stable diffusion backbones while remaining plug-and-play.

Different from prior approaches that optimize the initial latent, we consider the seed selection problem (Xu et al., 2024b; Li et al., 2025). Specifically, we perform early-stage screening of the seed pool, discarding those that are unlikely to yield faithful generations. This selection relies on the body token of the prompt—the core semantic anchor of the description—ensuring that only seeds aligned with the main concept are preserved, leading to more reliable and efficient T2I synthesis.

---

[1]Good seeds are initial latents that reliably yield high-fidelity, prompt-aligned, compositionally grounded, and globally coherent images; bad seeds fail to do so (See Figure 1).

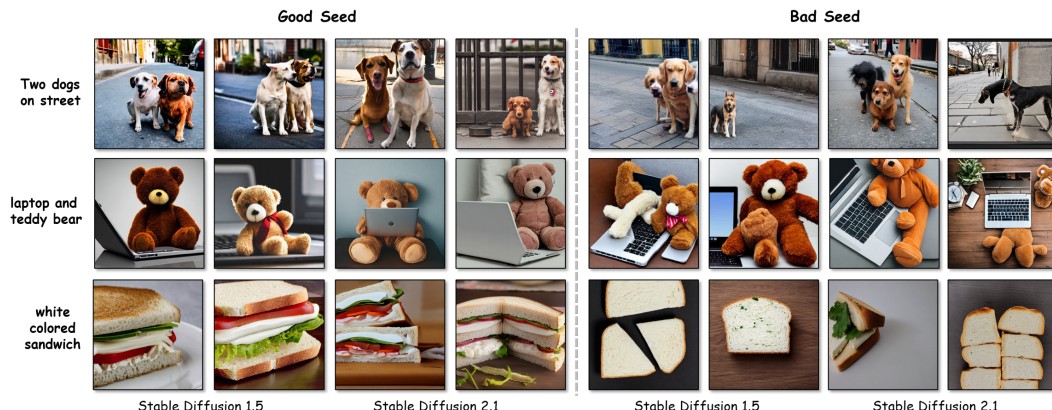

Figure 1: Illustrative examples of diffusion models initialized with good and bad random seeds. Good seeds typically generate coherent and faithful images aligned with text prompts, bad seeds may result in ghosting artifacts or images that are inconsistent with the intended description.

## 3 PRELIMINARIES AND MOTIVATIONS

In this section, we first introduce the preliminaries of stable diffusion model, illustrate the impact of seed, and then illustrate the motivations of our paper.

**Stable Diffusion**. Stable Diffusion (SD) (Rombach et al., 2022) is a widely used text-to-image generative model. Unlike conventional diffusion approaches that operate directly in pixel space (Ho et al., 2020; Nichol & Dhariwal, 2021; Dhariwal & Nichol, 2021; Song et al., 2021a;a), SD performs generation in the latent space of a pretrained autoencoder. This design enables high-resolution synthesis while substantially reducing computational cost, particularly during inference. Specifically, let $E(\cdot)$ denote an encoder mapping an image $x$ to a latent code $z = E(x)$, and let $D(\cdot)$ denote the corresponding decoder that reconstructs the image, such that $D(E(x)) \approx x$. After training the autoencoder, a Denoising Diffusion Probabilistic Model (DDPM) (Ho et al., 2020) is trained in this latent space. The DDPM iteratively denoises a noisy latent $z_t$ at each timestep $t$, conditioned on $c$, which in text-to-image scenarios corresponds to the text prompt. In SD, $c$ is produced by a pretrained CLIP text encoder (Radford et al., 2021) that maps the user's prompt to a vector. During training, clean latents are corrupted with Gaussian noise, and the denoiser is trained to predict the noise added at each timestep given the noisy latent, the timestep index, and $c$. This objective is implemented by minimizing the mean-squared error between the true and predicted noise. At inference time, an initial latent $z_T \sim \mathcal{N}(0, I)$ is sampled with the specific initial sample latent determined by a fixed seed s, where $T$ is the number of total timestipe. The denoiser then refines this latent step by step under the guidance of $c$ until finally a clean latent $z_0$ is obtained, which is then decoded by the pretrained decoder $D(\cdot)$ to produce the synthesized image $D(z_0)$.

**Impact of the Initial Seeds in SD**. A growing body of evidence (Xu et al., 2024b; Li et al., 2025) demonstrates that the initial seed—by fixing the starting noise—substantially steers the denoising trajectory and, in turn, the final image. Seeds induce systematic biases in outputs, affecting object arrangements (typical relative placement patterns), global style and tone, subject presence or absence, and even spurious artifacts. Consequently, the same text prompt can yield consistently good or poor results depending solely on the seed. As illustrated in Figure 1, for the same text prompt on SD 1.5 and SD 2.1, varying seeds produce outcomes that are visibly superior or inferior. This observation has motivated a line of work (Chen et al., 2024; Qu et al., 2023; Zheng et al., 2023; Couairon et al., 2023; Jia et al., 2024; Lian et al., 2024; Xu et al., 2023) that tackles generation quality from the seed side—often framed as seed optimization or seed-aware guidance.

**Motivations**. The motivation of this paper stems from two key aspects. First, as we illustrated above, seeds play a substantial role in shaping the quality of outputs in SD. Second, from a practical standpoint, a single text prompt typically requires multiple generated outputs for users to evaluate, compare, and select from. Consequently, there is a strong demand for a large pool of good seeds that can consistently produce high-quality images. Unlike prior studies on seed optimization or seed-aware training, this work focuses on the **seed selection** problem: given a collection of random seeds, how can we identify those that are likely to generate high-quality outputs?

# 4  ADSS: ATTENTION-DRIVEN SEED SELECTION

In this section, we present our method for seed selection in SD. A central challenge is that the quality of a seed, whether it is good or bad, can typically be assessed only after completing the entire diffusion process, which makes seed selection computationally expensive and logically problematic. To address this issue, we propose to monitor the diffusion process itself, with a particular focus on its early stages, to predict whether a given seed has the potential to generate desirable outputs. Specifically, we first analyze the evolution of attention patterns during denoising, which serve as a critical signal shaping the final image. Building on these observations, we then introduce our attention-driven seed selection method.

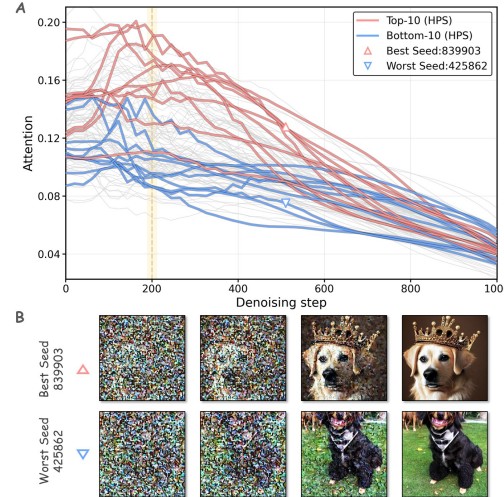

Figure 2: A. Trends of cross-attention on body token "dog" for 100 seeds throughout the denoising process. Red/blue curves represent high/low-quality outputs. Notably, as early as $t = 200$, good and bad seeds become roughly separable based on their cross-attention. B. Intermediate images of the best and worst seed.

## 4.1  OBSERVATION DURING DENOISING

Intuitively, the entire denoising process can be viewed as a form of continuous image optimization, where each intermediate image is iteratively refined. We therefore conjecture that a successful generation should begin by establishing the main outline—capturing the body tokens that correspond to the head noun(s) denoting the primary subject(s) expected to occupy the central visual mass—followed by progressive refinement of details. By body tokens, we specifically refer to the core subject terms (e.g., *cat* and *dog* in "*a cat and a purple dog*," or *turtle* in "*a fisheye lens view of a turtle in a forest*"), rather than modifiers or contextual descriptors. In other words, high-quality seeds should prioritize anchoring the dominant subjects before attending to secondary attributes and contextual information. By contrast, low-quality seeds disperse attention prematurely across modifiers and background tokens, which is associated with missing or undersized subjects, incorrect counts, and style-dominated artifacts.

Motivated by these insights, we consider seed selection by executing only a small number of denoising steps and evaluating whether the process prioritizes the body tokens early on. To achieve this idea, we leverage cross-attention (Guo et al., 2024; Chefer et al., 2023) on the body tokens as the criterion for early-stage seed selection. As illustrated in Figure 2, across diverse timesteps, good seeds (red curves) consistently allocate higher attention weights to body tokens from the very beginning of the denoising trajectory, whereas bad seeds (blue curves) fail.

## 4.2  METHOD

Figure 3 illustrates the framework of our proposed Attention-Driven Seed Selection (ADSS). ADSS is a lightweight and flexible seed selection method that can be seamlessly integrated into any existing SD pipeline. It identifies promising seeds by evaluating whether the intermediate images effectively capture the body tokens through the cross-attention mechanism. In the following sections, we first define the aggregated cross-attention map, and then describe how it is used to rank and select seeds.

**Aggregated Cross-attention Map**.  Text–image alignment in SD is achieved through cross-attention (Hertz et al., 2023), which integrates textual semantics into the denoising process. A text prompt $y$ is first encoded by CLIP into a conditional embedding $c$, formulated as $c = f_{\text{CLIP}}(y)$, where $c \in \mathbb{R}^{n \times d}$ with $n$ denoting the maximum token length and $d$ the embedding dimension. The conditional embedding $c$ is then linearly projected to obtain keys $K$ and values $V$, while queries $Q$ are derived from UNet activations. For a single cross-attention layer with one head and a seed $s$ sampled from the seed pool, the attention map is computed as

$$A^s = \text{softmax}\left(\frac{QK^\top}{\sqrt{d}}\right). \tag{1}$$

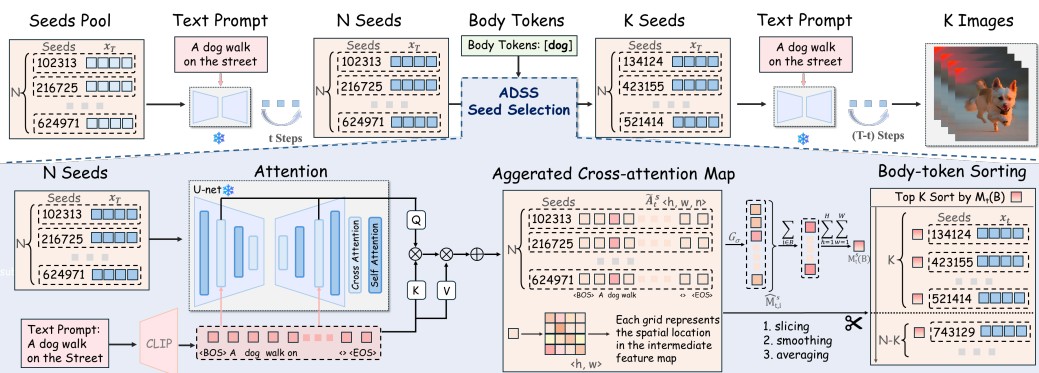

Figure 3: Framework of our proposed Attention-Driven Seed Selection (ADSS). The upper pathway illustrates the standard text-to-image generation pipeline with $N$ seeds. After a few denoising steps ($t$ steps), ADSS is applied to evaluate whether the intermediate feature maps effectively capture the body tokens at an early stage. It then selects the top-$K$ intermediate feature maps based on cross-attention and continues the remaining denoising steps with them.

Given a specific seed $s$, the element $A_t^s[h, w, i]$ represents the attention weight assigned to token $i$ among all tokens at spatial location $(h, w)$ in the intermediate feature map at timestep $t$. Higher values indicate stronger emphasis on token $i$.

Note that the UNet architecture in SD has multiple blocks with different resolutions (commonly $64, 32, 16, 8$), which leads cross-attention multi-head. To effectively aggregate prompt-conditioned attention, we define the **aggregated cross-attention map**: at a chosen spatial resolution with seed s, we collects cross-attention maps from the down/mid/up UNet blocks, stack all heads (and relevant blocks), reshape from the native (batch$\times$heads, $q$, $n$) format (with $q = h \cdot w$) to $(-, h, w, n)$, and average over the stacked dimension. This yields a single aggregated map $\bar{A}_t^s \in \mathbb{R}^{h \times w \times n}$. We then apply a temperature-scaled softmax along the token axis to sharpen the distribution,

$$\tilde{A}_t^s[h, w, i] = \frac{\exp\big(\beta\, \bar{A}_t^s[h, w, i]\big)}{\sum_{j=1}^n \exp\big(\beta\, \bar{A}_t^s[h, w, j]\big)},$$

where $\beta$ is usually set to be 100. By this means, for every spatial location $(h, w)$ we obtain a well-normalized probability distribution over tokens.

**Body-token Sorting: slicing, smoothing, averaging.** Given the aggregated attention map $\tilde{A}_t^s[h, w, i]$ for seed $s$, we focus on a subset of tokens indexed by $B$. Let $B \subseteq \{0, \ldots, n-1\}$ be the set of body-token indices under the original prompt indexing, including BOS/EOS (index 0 for BOS, $n-1$ for EOS). For each $i \in B$, we take the spatial slice, smooth it on the $(h, w)$ grid, and average it:

$$M_{t,i}^s[h, w] = \tilde{A}_t^s[h, w, i], \quad \widehat{M}_{t,i}^s = G_\sigma *_{\text{refl}} M_{t,i}^s.$$

Here $G_\sigma$ is a normalized $(2k+1) \times (2k+1)$ Gaussian kernel, and $*_{\text{refl}}$ denotes 2D discrete convolution with reflection padding at the boundaries. Pooling over space and average all body tokens gives the final body-token concentration at step $t$:

$$M_t^s(B) \;=\; \frac{1}{|B|\, H\, W} \sum_{i \in B} \sum_{h=1}^H \sum_{w=1}^W \widehat{M}_{t,i}^s[h, w].$$

Empirically, we use $M_t^s(B)$ in ADSS to pre-screen seeds. Consider the prompt "*a dog and a cat walking on the street*" (illustrated in Figure 3). Given a candidate pool $\mathcal{S} = \{s_1, \ldots, s_N\}$, where $N$ is the total number of seeds, we run only the first few denoising steps for each seed to obtain the aggregated cross-attention map $\tilde{A}_t^s[h, w, i]$, and then compute $M_t^s(B)$ for the body tokens dog and cat at the last of these early steps. We rank seeds by this score, retain the top-$K$ to complete the remaining generation, and discard the rest. For the top-$K$ seeds, we continue the remaining denoising steps and obtain $K$ output images.

Table 1: Mean at $k = 50$ on three datasets with four metrics (HPS, IR, PickScore, CLIP). For each metric, we bold the larger value between RANDOM, GOLDEN, and ADSS.

| Version | Method | DrawBench | | | | InitNO | | | | Pick-a-Pic | | | |
|---|---|---|---|---|---|---|---|---|---|---|---|---|---|
| | | HPS ↑ | IR ↑ | PickScore ↑ | CLIP ↑ | HPS ↑ | IR ↑ | PickScore ↑ | CLIP ↑ | HPS ↑ | IR ↑ | PickScore ↑ | CLIP ↑ |
| 1.4 | RANDOM | 0.2448 | -0.2192 | 20.5789 | 0.2561 | 0.2701 | 0.0025 | 21.7426 | 0.2729 | 0.2431 | -0.1648 | 20.2141 | 0.2561 |
| | GOLDEN | 0.2474 | -0.1852 | **20.6060** | 0.2568 | 0.2760 | 0.1249 | 21.8152 | 0.2745 | **0.2461** | **-0.1043** | **20.2748** | 0.2569 |
| | ADSS | **0.2481** | **-0.1774** | 20.6004 | **0.2581** | **0.2763** | **0.1512** | **21.8163** | **0.2758** | 0.2455 | -0.1318 | 20.2356 | **0.2573** |
| 1.5 | RANDOM | 0.2431 | -0.2547 | 20.5598 | 0.2559 | 0.2688 | -0.0428 | 21.7167 | 0.2729 | 0.2469 | -0.1381 | 20.2204 | 0.2568 |
| | GOLDEN | 0.2460 | -0.2248 | 20.5782 | 0.2558 | 0.2735 | 0.0153 | 21.7458 | 0.2723 | 0.2474 | -0.1311 | **20.2637** | 0.2568 |
| | ADSS | **0.2467** | **-0.2204** | **20.5784** | **0.2568** | **0.2742** | **0.0178** | **21.7582** | **0.2736** | **0.2479** | **-0.1301** | 20.2202 | **0.2571** |
| 2.0 | RANDOM | 0.2559 | 0.1398 | 21.0411 | 0.2683 | 0.2847 | 0.8185 | 22.2372 | 0.2916 | 0.2639 | 0.1583 | 20.5948 | 0.2652 |
| | GOLDEN | 0.2582 | 0.1730 | 21.0691 | 0.2689 | 0.2864 | 0.8317 | **22.2778** | 0.2922 | 0.2665 | **0.2489** | 20.6858 | 0.2672 |
| | ADSS | **0.2591** | **0.1911** | **21.0692** | **0.2703** | **0.2867** | **0.8324** | 22.2662 | **0.2925** | **0.2668** | 0.2285 | **20.6946** | **0.2675** |
| 2.1 | RANDOM | 0.2595 | 0.1609 | 21.1050 | 0.2629 | 0.2908 | 0.9992 | 22.3022 | 0.2918 | 0.2678 | 0.2587 | 20.7687 | 0.2695 |
| | GOLDEN | 0.2617 | 0.1877 | 21.1285 | 0.2636 | 0.2934 | 0.9384 | **22.4706** | 0.2922 | 0.2698 | **0.3014** | **20.8023** | 0.2694 |
| | ADSS | **0.2620** | **0.2018** | **21.1289** | **0.2645** | **0.2938** | **1.0023** | 22.4165 | **0.2929** | **0.2699** | 0.2722 | 20.7902 | **0.2698** |

## 5 EMPIRICAL ANALYSIS

We first introduce our experimental setting and conduct extensive experiments and evaluate the effectiveness and generality of ADSS through three guiding questions: **Q1: How does ADSS compare with baseline seed-selection strategies?** We benchmark ADSS against random seeding selection strategies and report the average of top-k performance. **Q2: Can ADSS serve as a plug-in to boost state-of-the-art seed or initial-noise optimization methods?** We use ADSS as a seed filter while keeping each downstream optimizer unchanged to assess additive gains. **Q3: Is focusing solely on body tokens sufficient to improve performance?** We ablate non-body tokens and test whether body-token-only yields consistent improvements without additional token-level modeling.

### 5.1 EXPERIMENTAL SETTING

**Datasets**. We evaluate on three prompt suites: *DrawBench*[2] (Saharia et al., 2022), a broad suite covering compositionality, commonsense, spatial relations, and fine-grained attributes; *InitNO* (Guo et al., 2024); and *Pick-a-Pic* (Kirstain et al., 2023), a large-scale human-preference–oriented benchmark used for alignment studies. These datasets jointly stress both semantic faithfulness and aesthetic preference, enabling a balanced assessment of text–image generation quality.

**Metrics**. We report total four preference-oriented metrics: HPS (v2.1) (Wu et al., 2023), IR (ImageReward), and PickScore (Kirstain et al., 2023), (Xu et al., 2024a)—trained on large-scale human-preference data and shown to correlate well with human judgments. We also include CLIP (Cherti et al., 2023; Schuhmann et al., 2022; Radford et al., 2021) text–image similarity as a proxy for semantic alignment between the prompt and the generated image. Higher values indicate better performance for all reported metrics.

**Implementation Details**. We use four Stable Diffusion backbones—SD 1.4 (CompVis, 2022), SD 1.5 (RunwayML, 2022), SD 2.0 (Stability AI, 2022a), and SD 2.1 (Stability AI, 2022b). Starting from SD 1.4 as a widely used 512 baseline, SD 1.5 improves prompt following and aesthetics. Transitioning to the 2.x line, SD 2.0 adopts an updated text encoder and supports higher native resolutions; SD 2.1 further tightens alignment and detail under the same recipe. For every backbone, all non–seed–related sampling hyperparameters—scheduler, guidance scale, and number of steps—are held fixed and identical across methods so that observed differences stem solely from the seed-selection strategy. For each text prompt, we keep the seed pool for our ADSS at 100, set the timestep to 200 for computing $M_t^s(B)$, and retain the top-$k = 50$. Metrics are computed as averages over these 50 generated images. We collect cross-attention maps from all layers and heads at a resolution of $16 \times 16$ pixels for SD 1.4/1.5 and $24 \times 24$ pixels for SD 2.0/2.1.

### 5.2 COMPARISON WITH BASELINE SEED SELECTION (Q1)

We compare against two seed-selection baselines to answer Q1. RANDOM sampling draws distinct seeds uniformly at random per prompt and uses them for generation. GOLDEN Seeds (Xu et al.,

---

[2]To improve readability, we format dataset names in italics and algorithm names in small caps.

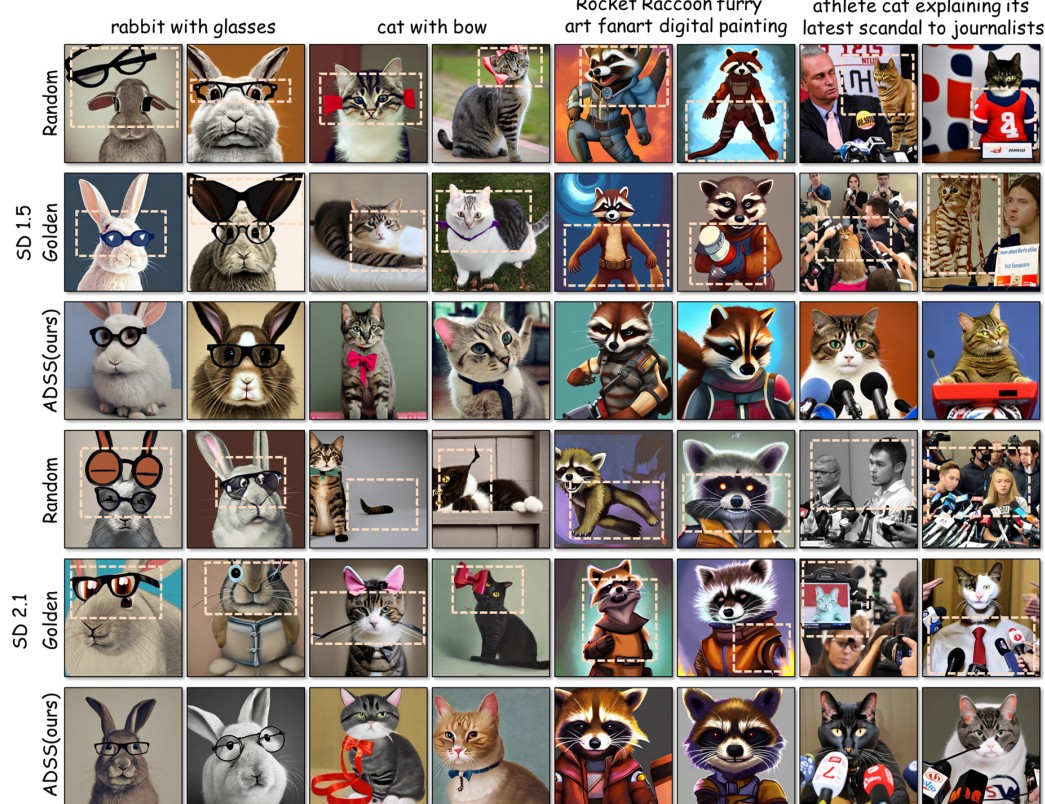

Figure 4: Qualitative comparative cases from three datasets. Images in the same column are generated from the same prompt. Orange dashed rectangles indicate problematic generated regions.

2024b) selects a small validation split of prompts per dataset, evaluates a candidate pool of seeds by their average HPS rank on the validation split, and then fixes the top $k$ seeds for all remaining prompts in that dataset. Table 1 presents the quantitatively comparative results and Figure 4 provides the generated images for three methods. Across SD 1.4/1.5/2.0/2.1 and three datasets, ADSS consistently outperforms RANDOM, with its largest, most reliable gains on IR and solid gains on HPS. On *DrawBench* with SD 1.4, IR rises from $-0.2192$ to $-0.1774$ (change 0.0418) and HPS rises from 0.2448 to 0.2481 (change 0.0033); PickScore is effectively tied. A similar pattern holds for SD 1.5 (IR increases by 0.0343, HPS by 0.0036). For SD 2.0/2.1 on *DrawBench*, ADSS attains the strongest mean at $k=50$ across metrics (e.g., SD 2.1: HPS 0.2620, IR 0.2018, PickScore 21.1289). On *InitNO* (SD 1.4/1.5), IR rises from 0.0025 to 0.1512 (change 0.1487) with consistent HPS gains.

Compared with GOLDEN, ADSS outperforms almost consistently on *DrawBench* and *InitNO*. On *Pick-a-Pic*, ADSS still improves HPS but margins shrink and IR, likely due to heterogeneous, highly compositional prompts (style/content/layout/post-processing) that spread a fixed $k=50$ seed budget across many competing modes and align better with a globally curated seed list. Importantly, where ADSS matches or surpasses GOLDEN, the win is meaningful: GOLDEN is a post-hoc, validation-driven global curation of good seeds reused across prompts, whereas ADSS is in-loop and per-prompt—scoring seeds at an early diffusion step, adapting without any validation split, adding

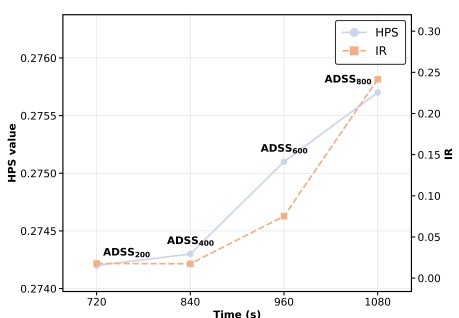

Figure 5: Performance and running time of ADSS on *InitNO* with different timesteps.

only $\sim$2.16 minutes per prompt with total 50 images —showing that early-step signals are predictive of final quality and that adaptive selection can rival or exceed post-processing–style seed curation while being simpler and more data-efficient.

We also present qualitative examples of generated images in Figure 4 using three methods—RANDOM, GOLDEN, and ADSS. As highlighted by the orange dashed boxes, it is common

for certain seeds to produce images whose content is incomplete, malformed, or focused on irrelevant objects. In contrast, ADSS substantially mitigates these failure modes, yielding images that align more faithfully with the intended textual descriptions.

To further validate ADSS based on the observations in Figure 2, we compare the ADSS seed-ranking list with the ground truth HPS ranking using (i) the overlap rate—the proportion of seeds common to both lists—and (ii) Normalized Discounted Cumulative Gain (NDCG) (Järvelin & Kekäläinen, 2002), which assesses how closely an ordering matches the ideal one while applying position-based discounts. ADSS achieves an NDCG of 0.93 and an overlap rate of 0.68, indicating that it effectively prioritizes valuable seeds and closely approximates the ground truth ranking.

We also investigate ADSS at different timesteps and observe that performance metrics generally improve as denoising progresses. As illustrated in Figure 5, HPS increases from 0.2742 to 0.2757 and IR from 0.0178 to 0.2416 when ADSS is evaluated at timesteps ranging from 200 to 800. This trend aligns with intuition: later denoising steps provide stronger signals for seed selection but incur higher computational costs. By selecting timestep $t = 200$, ADSS achieves a favorable trade-off between performance and efficiency, enabling practical early-stage seed selection.

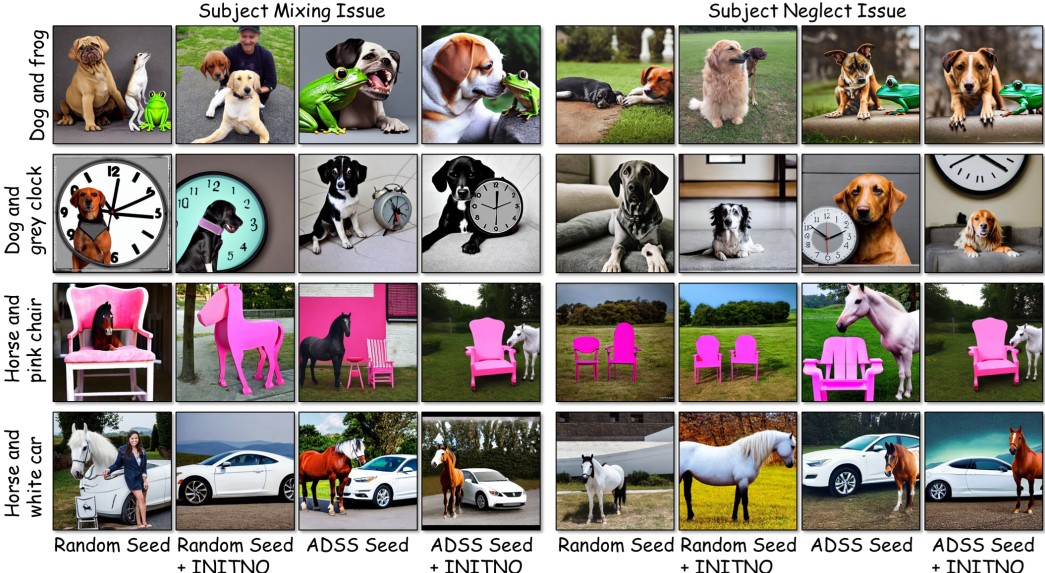

Figure 6: Comparison between random seeds and ADSS seeds for further seed optimization by INITNO to tackle the subject mixing and subject neglect issues.

## 5.3 PLUG-IN TO SEED/INITIAL-NOISE OPTIMIZATION (Q2)

In the previous section, we showed that ADSS is training-free and essentially zero cost, selecting high-quality seeds and outperforming other seed-selection methods. Here we test whether ADSS can be combined with state-of-the-art initial latent optimization to provide an extra boost. We consider two recent methods as baselines. INITNO (Guo

Table 2: Plug-in performance with ADSS selected seeds

| Method | Dataset | Version | HPS ↑ | IR ↑ | PickScore ↑ | CLIP ↑ |
|---|---|---|---|---|---|---|
| INITNO | *InitNO* | 1.4 | 0.2678 | -0.0510 | 21.6202 | 0.2671 |
| + ADSS | *InitNO* | 1.4 | **0.2701** | **-0.0062** | **21.6531** | **0.2696** |
| INITNO | *InitNO* | 1.5 | 0.2673 | -0.0524 | 21.6281 | 0.2672 |
| + ADSS | *InitNO* | 1.5 | **0.2689** | **-0.0083** | **21.6441** | **0.2686** |
| ZIGZAG | *DrawBench* | XL | 0.2947 | 0.7817 | 22.3633 | 0.2812 |
| + ADSS | *DrawBench* | XL | **0.2974** | **0.8026** | **22.4048** | **0.2817** |
| ZIGZAG | *Pick-a-Pic* | XL | 0.3125 | 1.0142 | 22.1336 | 0.2873 |
| + ADSS | *Pick-a-Pic* | XL | **0.3151** | **1.0344** | **22.1473** | **0.2878** |

et al., 2024) explicitly optimizes the initial noise per prompt under a chosen objective and then samples from the optimized initialization. ZIGZAG Diffusion Sampling (Bai et al., 2024) adds self-reflection to iteratively revise the sampling trajectory—including the starting latent—via alternating zigzag updates, improving prompt faithfulness and image quality without retraining. In our integration, ADSS serves only as a seed filter; baseline pipelines keep their default losses, schedulers, and hyperparameters—we simply replace uniform random seeds with ADSS-selected seeds.

Overall, plugging ADSS into both INITNO and ZIGZAG yields consistent improvements. For SD 1.4 with INITNO, ADSS raises semantic and preference metrics, with PickScore increasing by 0.0329

(21.6202 to 21.6531) and IR by 0.0448 (-0.0510 to -0.0062), while HPS and CLIP stay nearly unchanged. For SD XL with Zigzag on DrawBench, the largest gain is in PickScore, up by 0.0415 (22.3633 to 22.4048), alongside an IR increase of 0.0209 (0.7817 to 0.8026), with perceptual scores essentially stable. These examples indicate that ADSS supplies better starting latents and complements both optimization and trajectory-refinement pipelines.

In Figure 6, we show examples where, even after INITNO optimization, certain seeds still exhibit subject neglect (the primary subject does not appear) or subject mixing (two or more objects are conflated). For instance, in the horse–pink-chair case: without optimization the horse and chair are entangled; after optimization a pink "chair–horse" emerges; and in the missing-subject case the horse remains absent despite optimization. Our ADSS mitigates these failures by filtering such bad seeds, thereby solved such problem and strengthening INITNO or other pipelines.

## 5.4 ABLATION ON TOKEN TYPE (Q3)

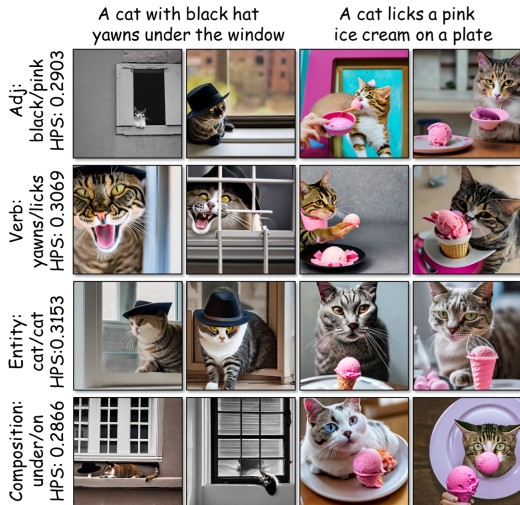

Figure 7: Examples of generated images when focusing on the different token types.

Beyond body tokens, we also investigate other token types to answer Q3. We ablate non-body tokens and test whether scoring based solely on body tokens yields consistent improvements without additional token-level modeling. For example, given the prompt "*A cat with black hat yawns under the window*," we categorize tokens as follows: *black* — adjective; *cat* — body tokens; *yawns* — verb; *under*— compostion. To test whether emphasizing body tokens outperforms variants that target adjectives or verbs, we run ADSS with alternative token subsets on *DrawBench*, which contains abundant adjective, verb, and body tokens. We compare three SD-1.4 variants: ADSSadj (adjectives only), ADSSverb (verbs only), and ADSS$_{body}$ (our default, body tokens). As shown in Table 3, focusing on body tokens yields the best overall performance, achieving higher HPS and better IR than using adjectives or verbs alone.

We also investigate the visualization effects of the generated images when ADSS focuson different type of tokens. Figure 7 shows that focusing ADSS on *verbs* (e.g., "yawns," "licks") encourages action-specific motion cues, but often omits or degrades key object details, yielding incomplete subjects and lower HPS. Focusing on *adjectives* (e.g., "black,"

Table 3: Performance of ADSS with different token types on *DrawBench* with SD 1.4

| Method | HPS ↑ | IR ↑ | PickScore ↑ | CLIP ↑ |
|---|---|---|---|---|
| ADSS$_{adj}$ | 0.2440 | -0.2374 | 20.8490 | 0.2617 |
| ADSS$_{verb}$ | 0.2438 | -0.2272 | 20.8345 | 0.2609 |
| ADSS$_{body}$ | **0.2473** | **-0.2195** | **20.8702** | **0.2619** |

"pink") propagates color/style broadly—bleeding into nearby objects and background—while the main body becomes less faithful. Emphasizing *composition* improves relative placement (window/plate context) but increases missing or fragmented foreground objects. In contrast, early focusing on *body tokens* preserves subject integrity and semantics, leading to the strongest HPS—answering Q3: ADSS only needs early attention on body tokens.

## 6 CONCLUSION

We revisited the seed effect in T2I and showed that early cross-attention to core tokens strongly predicts prompt alignment and image quality. Building on this, we proposed ADSS, a training-free, plug-and-play procedure that globally ranks seeds via a body-token concentration score and works across SD variants. Experiments on three benchmarks demonstrated consistent gains in faithfulness and visual quality, improving human preference and alignment metrics with minimal overhead. ADSS also serves as a lightweight preselection layer that complements seed-optimization and rollback/refinement pipelines. These results positioned attention-driven seed selection as a practical primitive for robust, cost-efficient T2I at scale.

## REPRODUCIBILITY STATEMENT

We release our code and implementation at `https://anonymous.4open.science/r/ADSS-11DE/README.md`. All experiments were conducted on a Linux server equipped with $40\times$ NVIDIA V100 (32 GB) GPUs. For experiments, we used the first 100 prompts of *Initno* and *drawbench*, and the first 50 prompts of *Pick-a-Pic*, as the validation set for GOLDENM; the remaining prompts were used to run experiments for the three methods.

## ETHICS STATEMENT

This study investigates seed selection for text-to-image diffusion and involves no interaction with human subjects and no collection of new personally identifiable information. All benchmarks used (e.g., *Initno*, *drawbench*, *Pick-a-Pic*) are publicly available and used under their respective licenses. We report implementation details to support reproducibility and mitigate misuse by filtering prompts that could produce harmful, unsafe, or discriminatory content. We do not train or fine-tune any models; thus computational resources and associated emissions are modest. We will release code to enable scrutiny and responsible reuse. Large language models were used solely to polish the writing of this paper, no use for the paper submission form, and for no other purpose. The authors declare no competing interests. We affirm adherence to the ICLR Code of Ethics.

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
