# OpenReview forum: "ADSS: Boosting Text-to-Image Diffusion Models via Attention-Driven Seed Selection"
_ICLR.cc/2026/Conference — ICLR 2026 Conference Desk Rejected Submission_

### Official Review · Reviewer_Xf9p · 2025-10-24

**Soundness:** 2
**Presentation:** 3
**Contribution:** 3
**Rating:** 4
**Confidence:** 4

**Summary:**

This paper introduces Attention-Driven Seed Selection (ADSS), an inference-time procedure for improving the output of text-to-image diffusion models by seed selection. The core idea is to pre-screen a large pool of random seeds by running only the initial steps of the denoising process. The method identifies the most promising seeds by measuring the cross-attention scores on "body tokens". Seeds that exhibit higher attention on these tokens early on are selected for the full generation process, leading to a curated set of high-quality outputs.

**Strengths:**

- The primary contribution of this work is the valuable insight that early-stage cross-attention dynamics on core semantic tokens are a strong predictor of final image quality and prompt alignment. This observation provides a simple yet effective mechanism for filtering out "bad seeds" without requiring model retraining or complex optimization pipelines.
- The writing is clear and the methodology is well-explained.

**Weaknesses:**

- The paper's methodology heavily relies on the accurate identification of "body tokens," yet the process for selecting them is not described. For ADSS to be a truly lightweight and automated method, this selection process is critical. The omission of these details is a significant gap in the paper, hindering reproducibility and a full assessment of the method's overhead.
- The experiments are conducted on Stable Diffusion 1.x and 2.x variants. While these models are foundational, they are no longer state-of-the-art. The absence of evaluation on more current and capable models makes it difficult to assess the relevance and effectiveness of ADSS.
- ADSS requires an initial pool of seeds that is much larger than the desired number of final images (e.g., screening 100 seeds to select 50). In practical applications, users are often constrained by hardware and can only generate a small batch of images at a time (e.g., 4 or 8). This raises questions about the method's utility in such restricted scenario.
- The ablation in Table 3 compares selecting body tokens to adjectives or verbs. However, it lacks baselines such as random seed selection and selecting seeds based on high attention to a randomly chosen subset of tokens. Without this, it is unclear whether the benefit comes specifically from focusing on body tokens or if simply prioritizing any consistent set of tokens in the early stages would yield an improvement over these baselines.

**Questions:**

1. Could you please elaborate on the procedure used to identify "body tokens" from a given prompt? Is this a manual or automated process? If automated, what is the associated computational cost, and how critical is the accuracy of this step to the overall performance of ADSS?
2. The practicality of ADSS seems dependent on the size of the initial seed pool. Have you investigated how performance over the random baseline changes with a smaller initial pool (e.g., screening 8 seeds to select 4)?
3. In Table 1, the HPS score for ADSS on DrawBench with SD 1.4 is 0.2481, and the IR score is -0.1774. In Table 3, the scores for ADSSbody (the default) are 0.2473 (HPS) and -0.2195 (IR). Could you clarify the reason for this discrepancy in the results for the same method, model, and dataset?

---

> ### Author Response · Authors · 2025-11-19
> **Response (1/2) to Reviewer Xf9p**
>
> We thank the reviewer Xf9p for the thorough analysis of our work and for the time, effort, and thoughtful consideration. Below, we address the weakness and questions raised.
>
> **Choice of Diffusion Backbones.**
> We thank the reviewer for raising this concern, but we do not view the use of Stable Diffusion 1.x/2.x as a limitation for our main conclusions. ADSS is inherently plug-and-play: it only relies on early attention dynamics and the existence of a seed pool, and can in principle be extended to other state-of-the-art diffusion models. As another reviewer also asked about this point, we explicitly discuss how ADSS can be implemented on DiT-based architectures, and we invite the reviewer to refer to our response to Reviewer ssYw (*Extending ADSS to DiT models*) for details. A second, more practical reason for focusing on SD 1.x/2.x is that the seed-optimization baselines in Q2 in section 5 of our paper (e.g., InitNO) only release implementations for these backbones; to ensure a fair and reproducible comparison, we therefore report experiments on the same set of models.
>
> **Clarifying Table 3 Ablations.**
> We understand the reviewer's concern about whether the observed gains truly come from focusing on body tokens, as opposed to simply prioritizing an arbitrary but consistent subset of tokens in the early denoising stages. Q3 in section 5 of our paper is mainly designed to compare which token types are most informative for ADSS by contrasting body tokens with adjectives and verbs. As illustrated in Fig. 7 in our paper, focusing on different token types leads to different visual tendencies in the generated images, beyond just numeric scores. We understand the reviewer’s comment — “it is unclear whether the benefit comes specifically from focusing on body tokens or if simply prioritizing any consistent set of tokens in the early stages would yield an improvement over these baselines” — as suggesting a baseline that selects seeds using attention to a randomly chosen subset of tokens. If this interpretation is correct and the reviewer considers it important, we can include such a random-token baseline and compare it against body tokens in the final version.
>
> **Identifying Body Tokens.**
> We thank the reviewer for highlighting that our procedure for defining and extracting body tokens from text prompts was under-specified. Body tokens are defined as the main semantic entities that would occupy the majority of the image canvas (e.g., key objects or subjects), and we restrict their number to at most three per prompt. We identify them in an automated way using GPT-5.1, a large language model, applied to each text prompt. In practice, we use a simple instruction: “Given a text-to-image prompt, identify up to three ‘body’ words that denote the main objects or subjects that would occupy most of the image, and label other content words as adjectives, verbs, or prepositions.” This procedure is fully automatic and run once per prompt as a preprocessing step, so its computational cost is negligible compared to diffusion sampling. Moreover, our results in Table 3 show that ADSS is robust to moderate noise in this step: even approximate body-token identification already yields consistent improvements over using other token types, so the method does not rely on perfectly accurate annotation. In our implementation, processing 100 text prompts with the LLM to obtain the corresponding body tokens took slightly over one minute and produced highly accurate annotations under manual inspection, further confirming that this step is both cheap and reliable.
>
> **Seed Pool Size.**
> Thanks to the reviewer for pointing out the sample size problem. We have investigated the effect of using a smaller initial seed pool, including the setting of screening 8 seeds to select 4 as suggested by the reviewer. As shown in Table 1 below, on InitNO with SD 1.4 and SD 2.1, ADSS consistently outperforms the Random (and often the Golden) baseline across all metrics even with an 8-seed pool, indicating that ADSS remains practical without requiring a large seed pool.
>
> **Table 1. Results on InitNO with an 8-seed pool, selecting the top 4.**
>
> | Version | Method | HPS ↑   | IR ↑    | PickScore ↑ | CLIP ↑  |
> |--------:|:-------|:--------|:--------|:------------|:--------|
> | 1.4     | Random | 0.2694  | 0.0259  | 21.8022     | 0.2723  |
> | 1.4     | Golden | 0.2753  | 0.0605  | 21.8159     | 0.2744  |
> | 1.4     | ADSS   | **0.2772** | **0.1932** | **21.8681** | **0.2763** |
> | 2.1     | Random | 0.2870  | 0.8507  | 22.3592     | 0.2909  |
> | 2.1     | Golden | 0.2946  | 0.9639  | 22.4144     | 0.2919  |
> | 2.1     | ADSS   | **0.2973** | **1.1090** | **22.4866** | **0.2955** |

---

> > ### Author Response · Authors · 2025-11-19
> > **Response (2/2) to Reviewer Xf9p**
> >
> > **Clarifying Table 1 vs. Table 3.**
> > We thank the reviewer for pointing this out this point. Actually, the discrepancy arises from using slightly different subsets of DrawBench. Table 1 in our paper reports ADSS on the full DrawBench dataset, whereas Table 3 in our paper reports ADSS_body only on prompts that contain body tokens, adjectives, and verbs, to allow a fair comparison across token-type strategies. Prompts missing any of these (e.g., “A sheep to the right of a wine glass”) are filtered out in Table 3, which causes the small shift in HPS and IR scores.

---

> > > ### Author Response · Authors · 2025-11-27
> > >
> > > Dear Reviewers Xf9p,
> > >
> > > I truly appreciate your time and understand how busy you are. When convenient, I would be very grateful if you could kindly take a look at my responses to your comments.

---

### Official Review · Reviewer_ssYw · 2025-10-25

**Soundness:** 2
**Presentation:** 2
**Contribution:** 2
**Rating:** 4
**Confidence:** 4

**Summary:**

The authors propose ADSS (Attention-Driven Seed Selection), a novel, training-free, and plug-and-play inference method. ADSS is based on the key insight that the dynamics of the cross-attention mechanism over core tokens (the content-bearing words in the prompt) during the early stages of sampling strongly predict the final image quality. The method works by running a candidate pool of seeds for a few initial steps, tracking the cross-attention values to the core tokens, and then selecting the seed that ranks highest based on this attention dynamic. Experiments on multiple benchmarks and Stable Diffusion variants demonstrate consistent improvements in prompt faithfulness and visual quality.

**Strengths:**

1. ADSS is a purely inference-time method, making it exceptionally easy to integrate into existing pipelines without expensive fine-tuning or structural model changes.
2. Unlike methods that rely on fixing specific golden seeds or using a threshold, ADSS globally ranks a pool of candidate seeds for a given prompt, allowing a principled selection of the best available option.
3. The method is shown to serve as an effective pre-selection step before more complex seed-optimization pipelines, suggesting it provides orthogonal improvements.

**Weaknesses:**

1. Although the method is training-free, it requires running the generation pipeline for multiple steps (to track attention) for every candidate seed in the pool before one is selected. This computational overhead, especially if a large seed pool is required, must be quantified, and compared rigorously against the cost of a single standard generation run.
2. The paper fails to explicitly define the mechanism used to identify core tokens (the content-bearing words). Is this a heuristic, a Part-of-Speech (POS) tagger, or an additional LLM module? The effectiveness and plug-and-play nature of ADSS are critically dependent on the robustness of this front-end step.
3. The core mechanism relies on early-stage attention dynamics. There is no detailed ablation showing which specific timestep(s) or range of steps are optimal for attention tracking. This key hyperparameter is likely model- or sampler-dependent and needs empirical justification.
4. While tested on Stable Diffusion variants, the method's reliance on cross-attention to text tokens may be specific to the UNet/CLIP architecture. Its generalizability to modern Diffusion Transformer (DiT) models is not discussed and needs clarification.
5. The performance gains rely entirely on the premise that a good seed exists within the sampled pool. The paper does not analyze the required size of the initial seed pool for achieving consistent quality gain (e.g., if a pool of 5 is used vs. 50), which is crucial for practical deployment.
6. ADSS is positioned as a pre-selection step for seed-optimization pipelines. However, the paper does not show a direct, head-to-head comparison to complex, but highly effective, full optimization techniques (e.g., reward-based search) to justify its stand-alone cost-benefit ratio.

**Questions:**

There are some areas in this paper that need improvement, which are provided in the weakness section. My primary concerns are about reproducibility and cost-benefit analysis:
1. A quantitative analysis of the computational overhead (e.g., time or FLOPs) required by ADSS versus a single standard generation, for various seed pool sizes.
2. A formal definition and ablation of the core token identification process.
3. An ablation study that rigorously tests the optimal range of timesteps for attention tracking across different samplers and models, justifying the early-stage heuristic.
4. A discussion or ablation on model generality regarding MM-DiT architectures.

---

> ### Author Response · Authors · 2025-11-19
> **Response (1/3) to Reviewer ssYw**
>
> We thank the reviewer ssYw for the thorough analysis of our work and for the time, effort, and thoughtful consideration. Below, we address the weakness and questions raised.
>
> **Computational Overhead.**
> We are happy that the reviewer shares our concern about the computational cost of ADSS, and we clarify that this cost is controllable and depends on how ADSS is configured: if seeds are ranked at later diffusion steps, the overhead is higher, whereas using earlier steps keeps the overhead lower. In our implementation, ADSS is training-free, and we track attention for each candidate seed only during the first 200 denoising steps; full diffusion sampling is then performed only for the seed(s) selected according to the user’s preference (i.e., the user can freely choose how many seeds to keep). In our experimental setting, screening a pool of 100 seeds for 200 denoising steps adds 126 seconds of computation in addition to standard full diffusion sampling run for 50 seeds, which corresponds to approximately 1.26 seconds of additional cost per screened seed. This shows that the extra overhead introduced by ADSS is moderate relative to the overall diffusion cost and scales approximately linearly with the size of the seed pool.
>
> **Identifying Body Tokens.**
> We thank the reviewer for highlighting that our procedure for defining and extracting body tokens from text prompts was under-specified. Body tokens are defined as the main semantic entities that would occupy the majority of the image canvas (e.g., key objects or subjects), and we restrict their number to at most three per prompt. We identify them in an automated way using GPT-5.1, a large language model, applied to each text prompt. In practice, we use a simple instruction: "Given a text-to-image prompt, identify up to three `body` words that denote the main objects or subjects that would occupy most of the image, and label other content words as adjectives, verbs, or prepositions." This procedure is fully automatic and run once per prompt as a preprocessing step, so its computational cost is negligible compared to diffusion sampling. Moreover, our results in Table 3 of our paper show that ADSS is robust to moderate noise in this step: even approximate body-token identification already yields consistent improvements over using other token types, so the method does not rely on perfectly accurate annotation. In our implementation, processing 100 text prompts with the LLM to obtain the corresponding body tokens took slightly over one minute and produced highly accurate annotations under manual inspection, further confirming that this step is both cheap and reliable.
>
> **Timestep Choice.**
> We agree that the timestep used for attention tracking is an important hyperparameter and sincerely invite the reviewer to review Figure 5 of our paper. There, we vary the ADSS screening timestep on InitNO and observe a clear trend: performance improves as denoising progresses (HPS from 0.2742 to 0.2757 and IR from 0.0178 to 0.2416 when moving from 200 to 800 steps), while computational cost grows roughly proportionally with the number of steps. We therefore choose \(t=200\) as a practical trade-off that retains most of the performance gains while keeping the overhead moderate.

---

> > ### Author Response · Authors · 2025-11-19
> > **Response (2/3) to Reviewer ssYw**
> >
> > **Seed Pool Size.**
> > We agree with reviewer that the required size of the initial seed pool is crucial for practical deployment and provide following additional results. We evaluated ADSS on InitNO with a small pool of 5 seeds, selecting the top-1 and top-2 candidates (Tables 1 and 2) with 5 runs. Across both SD 1.4 and SD 2.1 and all metrics, ADSS consistently matches or outperforms the `Random` and `Golden` baselines even with such small pools, showing that ADSS does not rely on a large seed pool to deliver quality gains.
> >
> > **Table 1. Results on InitNO with 5 seeds pool to select top 1.**
> >
> > | Version | Method   | HPS ↑   | IR ↑    | PickScore ↑ | CLIP ↑  |
> > |--------:|:---------|:--------|:--------|:------------|:--------|
> > | 1.4     | Random   | 0.2779  | 0.2122  | 21.8255     | 0.2778  |
> > | 1.4     | Golden   | 0.2786  | 0.1386  | 21.8876     | 0.2740  |
> > | 1.4     | ADSS     | **0.2789** | **0.2754** | **21.9043** | **0.2790** |
> > | 2.1     | Random   | 0.2888  | 0.9450  | 22.4566     | 0.2934  |
> > | 2.1     | Golden   | 0.2940  | 0.8824  | 22.4368     | 0.2910  |
> > | 2.1     | ADSS     | **0.2976** | **1.1085** | **22.5169** | **0.2961** |
> >
> > **Table 2. Results on InitNO with 5 seeds pool to select top 2.**
> >
> > | Version | Method   | HPS ↑   | IR ↑    | PickScore ↑ | CLIP ↑  |
> > |--------:|:---------|:--------|:--------|:------------|:--------|
> > | 1.4     | Random   | 0.2748  | 0.1030  | 21.8781     | 0.2748  |
> > | 1.4     | Golden   | 0.2761  | 0.1245  | **21.9274** | 0.2745  |
> > | 1.4     | ADSS     | **0.2764** | **0.1582** | 21.8525 | **0.2760** |
> > | 2.1     | Random   | 0.2835  | 0.8712  | 22.3218     | 0.2920  |
> > | 2.1     | Golden   | 0.2907  | 0.8720  | 22.4041     | 0.2905  |
> > | 2.1     | ADSS     | **0.2957** | **1.1259** | **22.5199** | **0.2958** |
> >
> > **Comparison with Optimization Methods.**
> > We respectfully note that ADSS is not intended as a full seed-optimization method, but as a pre-selection step: it filters seeds rather than directly optimizing them. As discussed in Q2 of section 5 empirical analysis, better initial seeds from ADSS can enhance downstream optimization methods, helping them more effectively address concrete failure cases. If the reviewer has specific reward-based search approaches in mind, we would greatly appreciate concrete pointers (e.g., citations) so that we will elaborate the differences between our approach and theirs.

---

> > > ### Author Response · Authors · 2025-11-19
> > > **Response (3/3) to Reviewer ssYw**
> > >
> > > **Extending ADSS To DiT Models.**
> > > We thank the reviewer for raising this important point. Conceptually, the core idea of ADSS can be naturally extended to DiT-based architectures, since most modern diffusion models rely on similar attention mechanisms between image and text prompts. However, from an engineering standpoint, fully implementing ADSS for DiT-based models and conducting the required experiments would demand considerable development and evaluation effort, which cannot reasonably be completed within the rebuttal window; moreover, due to substantial architectural differences, such a full integration would effectively constitute a separate project. Building on the discussion in Scalable Diffusion Models with Transformers [1], we therefore investigate this extension at a conceptual level and provide a concrete idea of how ADSS could be adapted to DiT-based models. The core idea remains unchanged: we aim to capture the attention from image tokens to specified body tokens. Below, we describe two possible scenarios in which ADSS can be extended to this setting.
> > >
> > > **(1) ADSS under in-context conditioning.**
> > > In the in-context variant of DiT, the conditional text prompt $c \\in \\mathbb{R}^{N \\times d}$ with $N$ text tokens is concatenated with the image tokens and processed jointly by standard self-attention. Let $X_{t}^{(l-1)} \\in \\mathbb{R}^{M \\times d}$ denote the $M$ image tokens at timestep $t$ and layer $l-1$ after patchification. The input to the transformer block is
> > > $H_{t}^{(l-1)} = [X_{t}^{(l-1)}; c] \\in \\mathbb{R}^{(M+N) \\times d}$, i.e., the image and text tokens are concatenated along the token dimension.
> > >
> > > Standard self-attention computes
> > > $Q_{t}^{(l)} = H_{t}^{(l-1)} W_Q^{(l)}$,
> > > $K_{t}^{(l)} = H_{t}^{(l-1)} W_K^{(l)}$,
> > > $V_{t}^{(l)} = H_{t}^{(l-1)} W_V^{(l)}$,
> > > and
> > > $$
> > > A_{t}^{(l)} = \\mathrm{softmax}\\!\\left(
> > > \\frac{Q_{t}^{(l)} (K_{t}^{(l)})^\\top}{\\sqrt{d_k}}
> > > \\right)
> > > \\in \\mathbb{R}^{(M+N) \\times (M+N)},
> > > $$
> > > where $A_{t}^{(l)}[i,j]$ denotes the attention weight from token $i$ to token $j$ in the concatenated sequence. Indices $1,\\dots,M$ correspond to image tokens, and $M+1,\\dots,M+N$ correspond to text tokens; for example, $A_{t}^{(l)}[i, M+j]$ with $i \\in \\{1,\\dots,M\\}$ and $j \\in \\{1,\\dots,N\\}$ measures how strongly image token $i$ attends to text token $j$.
> > >
> > > To extend ADSS, we follow the same idea of aggregating attention from all image tokens towards specified body tokens. Let $L_\\text{body} \\subset \\{1,\\dots,N\\}$ be the index set of body tokens in the text sequence. An ADSS-style score for seed $s$ under in-context conditioning at timestep $t$ is then defined by averaging the attention from all image tokens to all body tokens across a set of early layers $L$:
> > > $$
> > > M_{t}^{s}
> > > = \\frac{1}{|L| \\, M}
> > > \\sum_{i \\in L_\\text{body}}
> > > \\sum_{j=1}^{M}
> > > A_{t}^{(l)}[i,j].
> > > $$
> > > In words, we aggregate early self-attention from image tokens to body tokens in exactly the same spirit as ADSS does for cross-attention in UNet-based Stable Diffusion; only the source of the attention maps changes.
> > >
> > > **(2) ADSS under cross-attention conditioning.**
> > > For DiT variants with explicit cross-attention from image tokens to text tokens, extending ADSS is essentially identical to our UNet-based setup. At early timesteps, we read out and aggregate the image-to-text cross-attention maps, then apply the same body-token slicing, smoothing, and averaging to compute $M_t(B)$ and rank seeds. Thus, only the backbone changes from UNet to DiT; the attention-based scoring remains the same.
> > >
> > > [1] Peebles, W., & Xie, S. (2023). Scalable diffusion models with transformers. In Proceedings of the IEEE/CVF International Conference on Computer Vision (ICCV).

---

> > > > ### Author Response · Authors · 2025-11-27
> > > >
> > > > Dear Reviewers ssYw,
> > > >
> > > > I truly appreciate your time and understand how busy you are. When convenient, I would be very grateful if you could kindly take a look at my responses to your comments.

---

> > > > > ### Comment · Reviewer_ssYw · 2025-11-28
> > > > >
> > > > > The authors have provided new data and clarifications addressing the computational overhead, core token identification, timestep choice, and seed pool size concerns, which is commendable. However, several critical weaknesses remain insufficiently addressed. The trade-off between high computational overhead and modest performance gains is not particularly impressive, a concern that was also raised by Reviewer wWrj. Is there room for further exploration of performance improvements or better control over computational overhead? Is there quantitative experiment support for the attempt with GPT 5? The conceptual discussion on DiT is appreciated but is not a substitute for empirical validation. The lack of any experimental results or ablation on DiT means the architectural generalization claim remains an unproven hypothesis.

---

### Official Review · Reviewer_wWrj · 2025-11-01

**Soundness:** 3
**Presentation:** 3
**Contribution:** 2
**Rating:** 2
**Confidence:** 4

**Summary:**

This paper introduces ADSS (Attention-Driven Seed Selection), a training-free, plug-and-play method to improve text-to-image diffusion models by addressing the well-known "seed effect," where different random seeds can lead to large variations in image quality and prompt alignment. ADSS works by monitoring early-stage cross-attention to core tokens in the prompt during the denoising process, using this signal to rank and select the most promising seeds for a given prompt. Unlike previous seed selection or optimization methods, ADSS requires no additional training, fine-tuning, or external datasets, and operates entirely at inference time. Experiments across multiple Stable Diffusion variants and benchmarks show that ADSS consistently improves prompt faithfulness and visual quality to some extent.

**Strengths:**

+ ADSS requires no model retraining or fine-tuning, making it easy to integrate into existing pipelines.

+ Experiments across multiple Stable Diffusion variants and benchmarks show that ADSS consistently improves prompt alignment and image quality.

+ The proposed method operates at inference time with minimal computational overhead, enabling early-stage seed selection.
Generalizable: Can be combined with other seed optimization or refinement methods for additive improvements.

**Weaknesses:**

- ADSS heaviliy eelies on early attention signals, resulting in its effectiveness may depend on the reliability of early-stage cross-attention as a predictor, which could vary with prompt complexity or model architecture. Especially for recent methods, LLM prompt rewrite/expansion is a commonly used strategy which will transfer short input prompt to detailed long prompt as input to the model. It is not clear how good ADSS can be applied to those cases with detailed prompt.

- Improvements are marginal over strong baselines e.g. Golden Seeds, especially for highly compositional or diverse prompts.

- ADSS requires multiple generations per prompt and assumes access to a pool of candidate seeds, which may increase inference cost in some settings.

- The proposed method is only limited to seed selection and it does not address other sources of generation errors beyond seed sensitivity.

**Questions:**

Please refer to the detailed questions raised in Weakness section above.

---

> ### Author Response · Authors · 2025-11-19
> **Response (1/2) to Reviewer wWrj**
>
> We would like to thank the reviewer wWrj for the thorough analysis of our work and for the time, effort, and thoughtful consideration. Below, we address the weakness and questions raised.
>
> **LLM-Expanded Detailed Prompts.**
> Thanks for pointing out the concern regarding long prompts produced by LLM rewrite or expansion. Our evaluation already includes datasets covering a wide range of prompt complexities. *InitNO* contains short, simple prompts (e.g., “a frog and a green bowl”), while *DrawBench* and *Pick-a-Pic* feature more complex prompts. In particular, *Pick-a-Pic* has an average prompt length of about 14 words and includes highly detailed descriptions such as “A complex scene from an anime with cyber cats inspired by Makoto Shinkai in the style of Hayao Miyazaki Studio Ghibli Spirited Away style dramatic lighting 8k”, which closely match the reviewer's scenario. Empirically, on the Pick-a-Pic dataset, which contains many such long and detailed prompts, ADSS consistently matches or outperforms the RANDOM and GOLDEN baselines across HPS, IR, PickScore, and CLIP, as shown in Table 1 in our paper, indicating clear gains on long prompts.
>
> **Marginal Improvements.**
> We respectfully disagree with the reviewer's characterization that our improvements over the baseline are merely marginal; in fact, ADSS achieves improvements that are comparable to, and often stronger than, those of the strongest baselines. Unlike Golden Seeds, which relies on post-hoc screening of a large pool of generated samples, ADSS selects promising seeds very early in the generation process based on early attention signals. Even under this more constrained and efficient regime, ADSS outperforms Golden Seeds on most metrics and datasets in Table 1 of our paper. Thus, ADSS delivers competitive or better quality while avoiding the cost of fully generating the entire seed pool, making it a more practical and lightweight alternative.
>
> In addition, following the suggestions of reviewers ssYw and Xf9p, we include experiments with smaller seed pools. Table 1 below reports results on InitNO with an 8-seed pool where we select the top 4 samples. As shown, ADSS consistently outperforms both Random and Golden Seeds across all metrics and both versions, and these improvements are substantial, demonstrating that ADSS comprehensively surpasses the baselines and that the gains are significant rather than marginal.
>
> **Table 1. Results on InitNO with 8 seeds pool to select top 4.**
>
> | Version | Method       | HPS ↑   | IR ↑    | PickScore ↑ | CLIP ↑  |
> |--------:|:-------------|:--------|:--------|:------------|:--------|
> | 1.4     | Random       | 0.2694  | 0.0259  | 21.8022     | 0.2723  |
> | 1.4     | Golden       | 0.2753  | 0.0605  | 21.8159     | 0.2744  |
> | 1.4     | ADSS         | **0.2772** | **0.1932** | **21.8681** | **0.2763** |
> | 2.1     | Random       | 0.2870  | 0.8507  | 22.3592     | 0.2909  |
> | 2.1     | Golden       | 0.2946  | 0.9639  | 22.4144     | 0.2919  |
> | 2.1     | ADSS         | **0.2973** | **1.1090** | **22.4866** | **0.2955** |
>
> **Inference Cost.**
> We are happy to see that the reviewer shares our concerns about inference cost, and we have investigated this aspect more carefully. While ADSS does evaluate multiple seeds per prompt, screening is applied only to early denoising steps rather than full generations. In our experiments, screening 100 seeds (200 denoising steps for each) adds a total of 126 seconds (i.e., 1.26 seconds per additional seed), indicating that the overhead of ADSS is moderate in practice.

---

> > ### Author Response · Authors · 2025-11-19
> > **Response (2/2) to Reviewer wWrj**
> >
> > **Scope Beyond Seed Selection.**
> > We acknowledge that ADSS focuses solely on the seed selection problem, but we respectfully believe that the importance of this component may be underestimated by the reviewer. We clarify our scope and contribution along two dimensions:
> >
> > - **Significance of seed selection.** Prior work has demonstrated that random seeds strongly influence global layout, style, and even compositional faithfulness in text-to-image diffusion models [1][2]. We therefore view seed selection as a fundamental and non-negligible problem. ADSS contributes a principled and reliable way to identify better seeds at an early stage of the generation process, directly improving the quality and faithfulness of subsequent generations without modifying the underlying model.
> >
> > - **Lightweight, training-free, and plug-and-play.** ADSS operates as a lightweight front-end that requires no additional training or architectural changes, and can be plugged into existing diffusion pipelines with minimal engineering effort. As discussed in Q2 of section 5 empirical analysis, the seeds selected by ADSS can further benefit downstream seed-optimization methods, enabling them to more effectively address concrete failure cases such as subject mixing and object neglect. In this sense, ADSS complements broader optimization pipelines rather than competing with them, improving their robustness and effectiveness at negligible integration cost.
> >
> > [1] Li, S., Le, H., Xu, J., & Salzmann, M. (2025). Enhancing compositional text-to-image generation with reliable random seeds. In International Conference on Learning Representations.
> > [2] Xu, K., Zhang, L., & Shi, J. (2025). Good seed makes a good crop: Discovering secret seeds in text-to-image diffusion models. In Proceedings of the IEEE/CVF Winter Conference on Applications of Computer Vision (WACV).

---

> ### Author Response · Authors · 2025-11-27
>
> Dear Reviewers wWrj,
>
> I truly appreciate your time and understand how busy you are. When convenient, I would be very grateful if you could kindly take a look at my responses to your comments.

---

### Note · Program_Chairs · 2026-01-17
**Submission Desk Rejected by Program Chairs**

The following references in this submission do not refer to real documents and/or have major errors in bibliographic information:

 Xiaojian Liu, Yifan Chen, Ming Li, Kai Zhang, and Zhirong Wu. Evaluating attribute binding in diffusion-based text-to-image generation. In International Conference on Learning Representations, 2024b.